# Reconstructing single-cell resolution from spatial transcriptomics with CellRefiner

Eric Bourgain-Chang[1,7], Xiangyu Kuang ®[2,3,7], Zixuan Cang ®[4,5] ✉ & Qing Nie ®[2,3,6] ✉

Single-cell RNA sequencing technologies profile the transcriptome of individual cells but lack the spatial context necessary for dissecting cellular interactions like cell-cell communications. On the other hand, most current spatial transcriptomic technologies lack cellular resolution, limiting their capability for realistic downstream analysis. Here we present CellRefiner, a physical model-based method that integrates a single-cell dataset with a paired spatial dataset to generate single-cell resolution in the imputed spatial data. CellRefiner models cells as particles connected by forces, and then optimizes cell locations with spatial proximity constraints, gene expression similarity, and ligand-receptor interactions between cells. We systematically benchmark CellRefiner over a variety of simulated and real datasets using Visium, MERFISH, seqFISH, Slide-seqV2, and STARmap datasets to demonstrate its accuracy, robustness, and ability to recover spatial patterns of cells. We also demonstrate its utility for improving spatially dependent analysis over the original spatial data for the contact-based cell-cell communication on mouse cortex and lymph node tissues. Our results show CellRefiner is capable of reconstructing single-cell resolution from non-single-cell resolution spatial data, allowing downstream analysis that requires individual-cell resolution and spatial information.

Single-cell RNA sequencing (scRNA-seq) enables profiling individual cells at high-resolution and high-throughput, enabling new analyses of gene expression heterogeneity and interactions[1,2]. However, scRNA-seq methods require dissociation of the tissue and subsequent loss of spatial information, which is critical to studying biological processes such as spatially constrained cell-cell communications[3,4]. In contrast, spatial transcriptomics (ST)[5,6] retain the spatial context necessary for investigating spatial gene expression patterns and organization of cells. However, those techniques have their own shortcomings, limiting their potential for downstream analyses. Image-based approaches such as seqFISH+[7] and

MERFISH[8] do not cover the whole transcriptome while sequencing-based approaches such as Visium[9] and Slide-seqV2[10] do not preserve individual cell identity, with spatial resolution in a spot that contains a small number of cells. To facilitate analyses that require both single-cell resolution and large gene coverage, computational methods are needed to integrate scRNA-seq and ST data.

Currently, several computational methods have been developed to assign spatial context to scRNA-seq data. Many of these methods integrate scRNA-seq with ST data and deconvolve spatial spots into proportions of cell types that are determined in scRNA-seq data, such as Cell2location[11], SpatialDWLS[12], SPOTlight[13], and DSTG[14]. Another

[1]Mathematical, Computational, and Systems Biology PhD Program, University of California, Irvine, Irvine, CA 92697, USA. [2]Department of Mathematics, University of California, Irvine, Irvine, CA 92697, USA. [3]NSF-Simons Center for Multiscale Cell Fate Research, University of California, Irvine, CA 92697, USA. [4]Department of Mathematics, North Carolina State University, Raleigh, NC 27695, USA. [5]Center for Research in Scientific Computation, North Carolina State University, Raleigh, NC 27695, USA. [6]Department of Developmental and Cell Biology, University of California, Irvine, Irvine, CA 92697, USA. [7]These authors contributed equally: Eric Bourgain-Chang, Xiangyu Kuang. ✉e-mail: zcang@ncsu.edu; qnie@uci.edu

class of approaches estimates spatial proximity between cells in scRNA-seq based on biological assumptions. CSOmap[15] constructs a cell-cell affinity graph using contact-based ligand-receptor interactions, although the lack of spatial reference may result in false positives. SPROUT[16] similarly uses ligand-receptor interactions, along with a deconvolution method to construct an affinity matrix, which uses a low-dimensional space to reconstruct a single-cell resolution quasi-structure. novoSpaRc[17] assumes a correlation between gene expression similarity and spatial closeness and uses optimal transport to construct a probabilistic matching of scRNA-seq data to a predefined grid. While these methods can reveal cell type heterogeneity and composition in ST data, the lack of individual cell resolution in the method limits their utility for recovering biological processes, such as cell-cell communication at individual cell level[18–20].

There has been an increasing interest in recovering single-cell resolution of non-single-cell ST[21]. Seurat[5] uses canonical correlation analyses to embed scRNA-seq and spatial data to a common latent space then project scRNA-seq cells to spots in the spatial data. Tangram[22] uses a deep learning framework to align single-cell data in space, creating a cell-to-spot mapping. While these probabilistic mappings can assign spatial locations to cells, they are mainly used for label transfer and cell type deconvolution. More recently, some methods start to focus on assigning spatial locations to individual cells, including CellTrek[23] that uses a co-embedding approach with a random forest model and CeLEry[24] that trains a neural network to predict spatial location from gene expression. These methods, however, place the cells independently which may cause unrealistic spatial arrangement of cells, such as overlapping, overly crowded, or overly sparse in cell distribution.

To address those problems, we utilize a physical model drawn from subcellular element model originally developed for simulating multicellular organization of cells, such as cell proliferation, cell-cell adhesion, and cell-cell communication[25]. Such an approach has been widely used for simulating large numbers of cells, with cells represented as elastically linked particles interacting via short-range potentials[26]. In such a model, each particle tracks a fully independent state with individual parameters that reflect heterogeneity, while known cell-cell interactions and single-cell dynamics can be directly

implemented as forces on particles[27]. For example, it can be used to simulate platelet-blood flow interactions[28] with platelets acted on by three forces corresponding to the effects of stochastic cell movement, intercellular interactions such as ligand-receptor binding, and fluid-cell interactions. Such a model can replicate experimental observations by directly relating parameters to experimental values while achieving linear scaling in computational efficiency[28]. These methods are valuable for computationally efficiently modeling thousands of cells subject to multiple types of interactions in a noisy environment.

Here, we present CellRefiner, a physical model-based method that can reconstruct single-cell resolution by utilizing both non-single-cell spatial transcriptomic data and scRNA-seq data. CellRefiner uses forces based on gene expression similarity and ligand-receptor interaction to systematically optimize cell locations modeled as particles. This approach naturally enforces spacing among cells while accounting for local spatial configurations resulting from cells with similar gene expression and ligand-receptor signaling between cells. The method can be applied to Visium, MERFISH, seqFISH+, Slide-seqV2, and STARmap[29] datasets, uncovering spatial organization of cells at single-cell resolution. In addition, we demonstrate the utility of CellRefiner for improving inference on cell-cell communication.

## Results
### Overview of CellRefiner

CellRefiner integrates scRNA-seq and ST data and predicts a high-fidelity single-cell resolution spatial data by using the subcellular element method, a particle-based model (Fig. 1). First, a mapping matrix based on the gene expression similarity between cells in scRNA-seq data and spots in ST data is generated to initialize the model (Fig. 1a). This initialization is adopted from Tangram[22] with additional regularization for spatial smoothness. CellRefiner is also compatible with other mapping methods for this task which can replace the initialization module. The mapping matrix is then used to assign several cells in scRNA-seq data to each spot in ST data, for example, 5 to 10 cells to each spot in Visium data. The assigned cells are initially randomly placed in an area covered by the corresponding spot. While the initial random placement of single cells can facilitate identification of large-

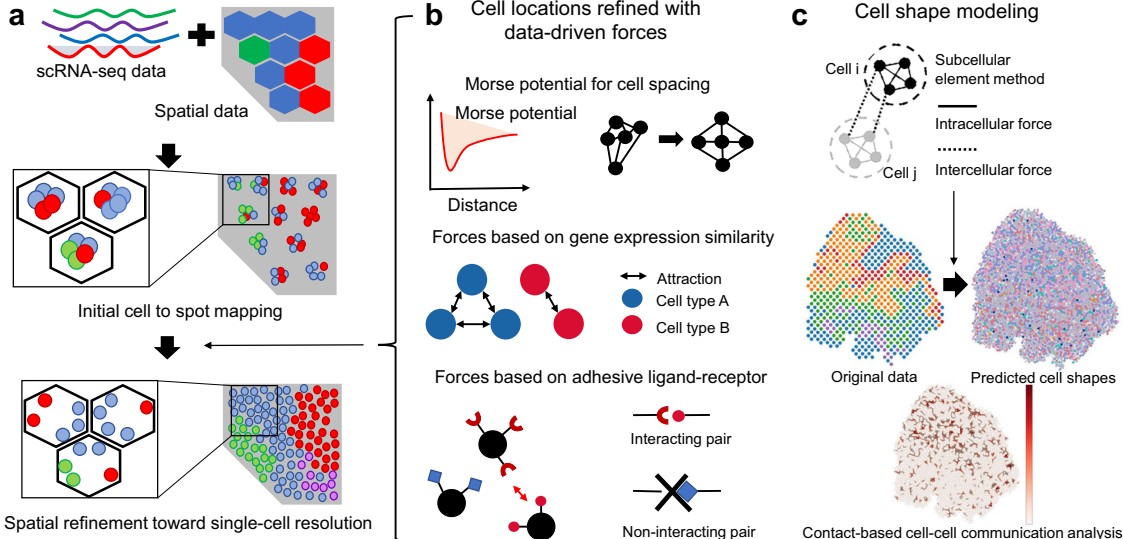

**Fig. 1 | Overview of CellRefiner. a** CellRefiner is a two-step method that first maps cells from scRNA-seq data to spot spatial regions in paired ST data, then refines the spatial locations of the mapped cells to generate single-cell resolution data. **b** The cell to spot mapping is formed by maximizing the cosine similarity between the gene expression matrices of the two datasets. Cells are assigned to the spatial region of the spot with the most similar gene expression to initialize the iterative particle-based model composed of three forces: 1) an elastic force from the Morse

potential spaces particles according to desired cell density and computational convergence; 2) particles are attracted to each other proportionately to the similarity of their corresponding cells' gene expression; 3) ligand-receptor affinity based on adhesive ligand-receptor interactions attract pairs of particles to each other. **c** After spatial refinement, CellRefiner models cell shapes using the subcellular element methods, enabling contact-based cell-cell communication analysis at spatially resolved single-cell resolution.

scale spatial patterns, it is not sufficiently accurate for analyzing fine-scale patterns and processes at single cell level. Therefore, we introduce a particle-based model to fine-tune the spatial locations of the single cells (Fig. 1b). In this model, cells are modeled as particles whose movements are guided by several different forces. The model contains three major components. First, a uniform Morse potential is applied to every pair of cells to recover the realistic distance between cells[26]. This force prevents the overlapping of cells and formation of large voids in tissues. Second, an attraction force is applied between cells based on gene expression similarity, with an assumption that cells with similar gene expression profiles are likely spatially close[30–32]. Third, based on known ligand-receptor interactions as grouped in CellChatDB, another attraction force is applied to cells expressing compatible ligand-receptor pairs[15,16,33,34]. In ablation analysis, incorporating LR forces leads to only marginal improvement, with performance nearly identical to that obtained without LR forces. We therefore treat the LR term as optional. Furthermore, CellRefiner models detailed cell shapes by representing each cell with multiple particles using the subcellular element method, facilitating the contact-based cell-cell communication analysis (Fig. 1c). Together, CellRefiner introduces a versatile model that incorporates the important effect of spacing between cells, gene expression similarities, and ligand-receptor interactions to produce a more biologically realistic single-cell resolution spatial transcriptomics data.

## Benchmarking spatial refinement performance on single-cell resolution ST data

We first validate the location refinement of individual cells with given ground truth mappings between scRNA-seq cells and ST spots. We generate pseudo multi-cell resolution ST data, named as pseudo-Visium data to resemble the resolution of Visium data, from a MERFISH dataset of the mouse hypothalamic preoptic region[35] (Fig. 2a). A grid of pseudo spots is placed on the tissue region and a cell in the original MERFISH data is assigned to a pseudo spot if the cell is covered by the spot (Supplementary Fig. 1). Here the cell-to-spot mapping is known due to the pseudo-Visium data construction, and we then evaluate the method given the accurate initial mappings to construct spatial data at the resolution of the original MERFISH data (Fig. 2a). In this sample, the ependymal cells form a well-defined and fine-scale structure that we use to assess the effectiveness of our method. Each cell is initially placed according to a normal distribution at its corresponding spot's center. The initial locations are then further perturbed proportionally to the spot radius to evaluate the performance of CellRefiner when handling noisy initialization (Fig. 2b). Kernel density estimation with Gaussian kernels[36] is used to estimate the spatial densities (Fig. 2b). For systematic evaluation, we use KL divergence between the spatial density of each cell type from CellRefiner and from ground truth data (Fig. 2c). The KL divergence is robustly reduced through CellRefiner, and the predicted spatial locations of cells are found to significantly outperform the random initial placement from the ground truth mapping (Supplementary Fig. 2). We further evaluate the smoothness of neighborhood structure by measuring distance from each cell in the output to its nearest neighbor of the same cell type in the output. A significant improvement in post refinement is observed when benchmarked with the neighborhood structure of the ground truth data (Fig. 2c). This improvement is qualitatively visible across cell types as well for the entire dataset, particularly for the ependymal cells (Fig. 2c).

Having demonstrated CellRefiner given a known mapping, next we test the entire method using a pseudo-Visium dataset constructed from various datasets without a known cell-to-spot mapping. Using a MERFISH data[35], we construct pseudo-Visium to form two datasets with a known ground truth where the gene expression of cells assigned to a pseudo spot is added as representation of the spot. Starting from the single-cell gene expression and pseudo-Visium data, we reconstruct the original data and compare it with the MERFISH data (Fig. 2e).

In particular, we observe that CellRefiner outputs the same distinct spatial structures as well as distribution of cell types (Fig. 2e). Repeating this process on a mouse embryo seqFISH[37] dataset with many cells also gives similar results between the output and the original (Fig. 2e). On Slide-seqV2 and STARmap datasets for the mouse hippocampus and visual cortex, respectively, CellRefiner produces simulations with distinct tissue structures visible such as the subiculum and layers of the cortex (Fig. 2e). These two datasets have substantially different geometries and numbers of cells, demonstrating CellRefiner's versatility over different dataset types.

Using single-cell resolution datasets as ground truth, we benchmark the performance of CellRefiner's ability to spatially localize cells relative to existing methods for mapping cells to space. We start with MERFISH data from the mouse hypothalamic preoptic region with a distinctive structure of ependymal cells (Fig. 2d). From this single-cell resolution data, we create synthetic spot data to form a paired dataset and map cells to space using CytoSPACE, Tangram, and CellTrek along with the corresponding outputs from CellRefiner (Fig. 2d). CellRefiner reduces the error compared to the ground truth for all three methods using two different metrics across all cell types (Fig. 2d). The Wasserstein distance penalizes individual cells mislocated more heavily than the KL divergence, which is calculated over a Gaussian distribution fit to each cell type's spatial configuration. We compare again with a mouse hippocampus Slide-seqV2 dataset consisting of several times as many cells and genes as the previous dataset (Fig. 2d). We create synthetic spot data and input the synthetic data and original gene expression data into CytoSPACE, Tangram, and CellTrek (Fig. 2d). In particular, the spatial regions marked by cell types such as the dentate pyramidal cells and endothelial cells are mapped more distinctly. However, other cell types have less defined spatial distributions, so we see the most improvement from CellRefiner when assessing with the Wasserstein distance (Fig. 2d). When we isolate one of these labeled cell types, there is a distinct structure following from the CA3 to the CA2, CA1, and subiculum along with a scattering of cells around the sample (Fig. 2d). We quantify this spatial dispersion using the Ripley's L, which shows CellRefiner produces values closer to the ground truth when applied to each method (Fig. 2d). In addition, we examined the performance of CellRefiner on cell types with different numbers of cells. We also compared CellRefiner with CeLEry using pseudo-Visium data generated from a Xenium dataset of human breast cancer[38], applying both methods to the paired pseudo-single-cell (cells taken from Xenium without spatial information) and pseudo-Visium spots. Under this setup, CellRefiner outperforms CeLEry, whereas CeLEry achieves higher accuracy when applied in a supervised setting using ground-truth Xenium spatial information (Supplementary Fig. 3). These results highlight the strength of CellRefiner for mapping single cells to multi-cell-resolution ST data. Lastly, while CellRefiner performs consistently well on various cell type sizes, its performance decreases on low-abundance cell types that composite about less than 5% of the total cell population (Supplementary Fig. 4).

## CellRefiner reveals scaled cellular structure in multi-cell resolution ST data

Now we examine the utility of CellRefiner when applied to paired scRNA-seq and multi-cell resolution ST datasets, particularly a 10X Visium dataset of the mouse cortex (Fig. 3a) with paired scRNA-seq data[39]. The CellRefiner output shows a clear spatial layer trend using the annotations from the scRNA-seq data (Fig. 3b). When used in the domain segmentation, where SpaceFlow is applied to the CellRefiner reconstructed single-cell resolution spatial data, the clear spatial layer trends are observed (Fig. 3c). This is quantitatively demonstrated using descriptive statistics of the spatial patterns of cell types, in particular Ripley's L function[40] for showing that both the original Visium and the CellRefiner data exhibit similar levels of dispersion (Fig. 3d). Another spatial analysis is to calculate the neighborhood enrichment score based on proximity

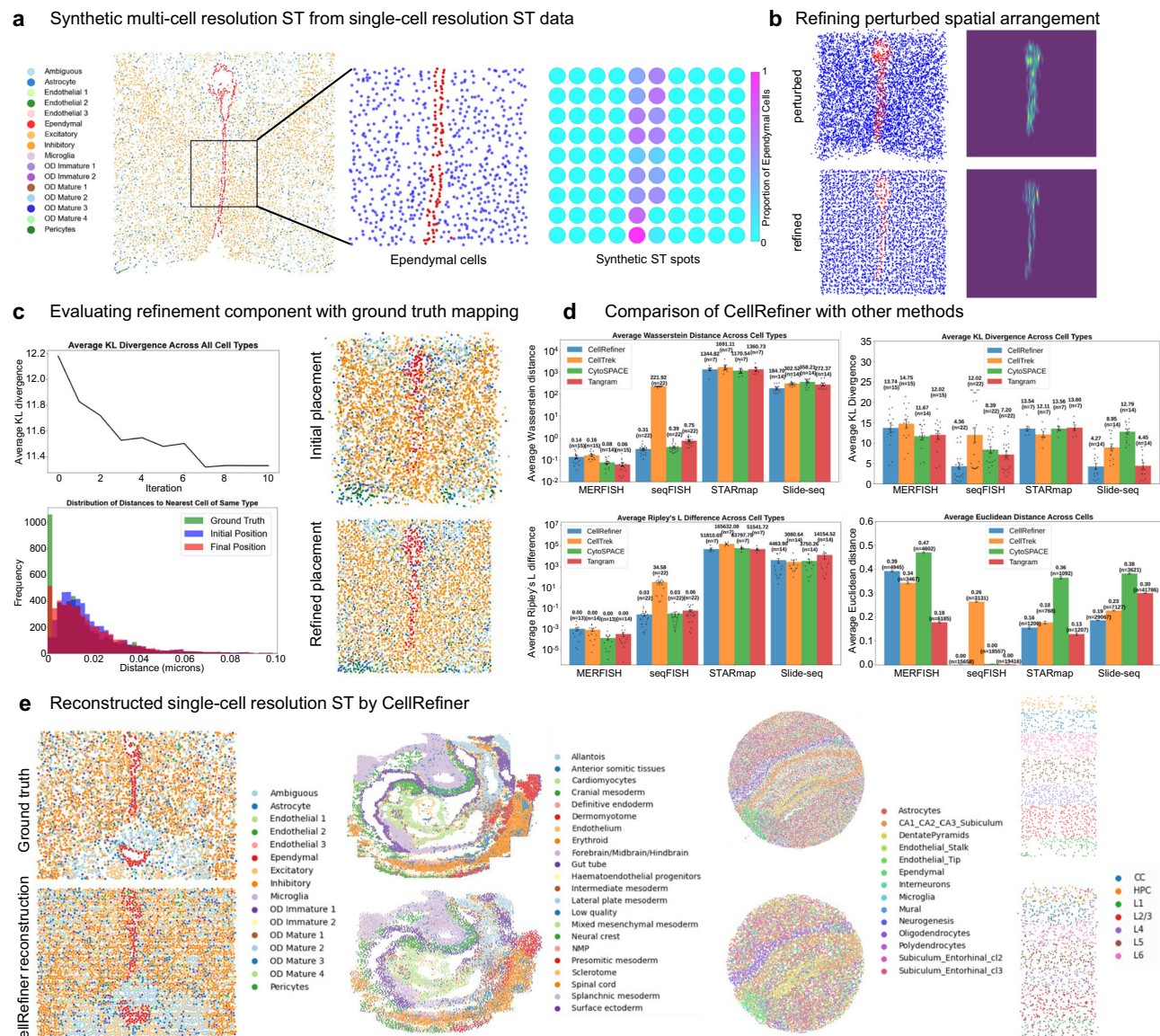

**Fig. 2 | Evaluation of CellRefiner with or without ground-truth mapping on single-cell resolution data. a** MERFISH slice from the mouse hypothalamic preoptic region used as ground truth for benchmarking CellRefiner using a known cell-to-spot mapping. Zoomed-in view shows a region of the tissue structure around the ependymal cells (yellow) for creation of simulated data. Cells are aggregated to form pseudo-Visium spot data, which is used to initialize the physical model, followed by the spatially refined output. **b** Spatially perturbed MERFISH data before and after CellRefiner processing, with ependymal cells in red, and density estimation of the first-time step and last time step of the refinement process. **c** Refinement error calculated as KL-divergence between density estimations of CellRefiner output and ground truth over simulation iterations for ependymal cells.

**d** Performance comparison of spatial mapping methods using multiple metrics. Comparison of CellRefiner, CellTrek, CytoSPACE, and Tangram on four single-cell resolution spatial datasets. Bar heights represent mean values of metrics with error bars representing standard error of the mean. For the three metrics across cell types, each dot represents one cell type and n indicates the number of cell types. For Euclidean distance across cells, *n* indicates the number of cells. Source data are provided in the Source Data file. **e** Reconstructed single-cell resolution spatial data by CellRefiner of MERFISH, seqFISH, Slide-seqV2, and STARmap data.

of the connectivity graph[41,42] (Fig. 3e). This is useful for downstream tasks like identifying spatially variable genes or candidates for cell-cell communication[42]. Applying CellRefiner to a paired Visium and scRNA-seq murine auricular lymph node dataset[43] (Fig. 3f) shows the tissue structure consistent with microscopy images[44,45], with higher concentrations of CD8 T cells around the T cell zone, whereas the Mature B cells are more concentrated towards the outer regions (Fig. 3g).

## CellRefiner provides insight into contact-based cell-cell communication

One important form of cell-cell communication is the contact-based signaling between cells with physical contact through membrane-

bound ligand and receptors. To investigate contact-based communication, both single-cell resolution gene expression information and spatial locations are needed. We first evaluate cell shape reconstruction by CellRefiner on single-cell resolution ST data. In the seqFISH+ data of mouse cortex[7], the reconstructed cell shapes qualitatively recapitulate the ground truth shapes and reliably reconstruct contact maps between cells, which are crucial for the contact-based cell-cell communication analysis (Fig. 4a). When applied to a seqFISH dataset of mouse embryo, the reconstructed shapes allow geometrically realistic analysis of cell contacts (Fig. 4b). Finally, we applied the full CellRefiner pipeline with each cell modeled by multiple particles to the multi-cell resolution Visium data. The reconstructed single-cell resolution data

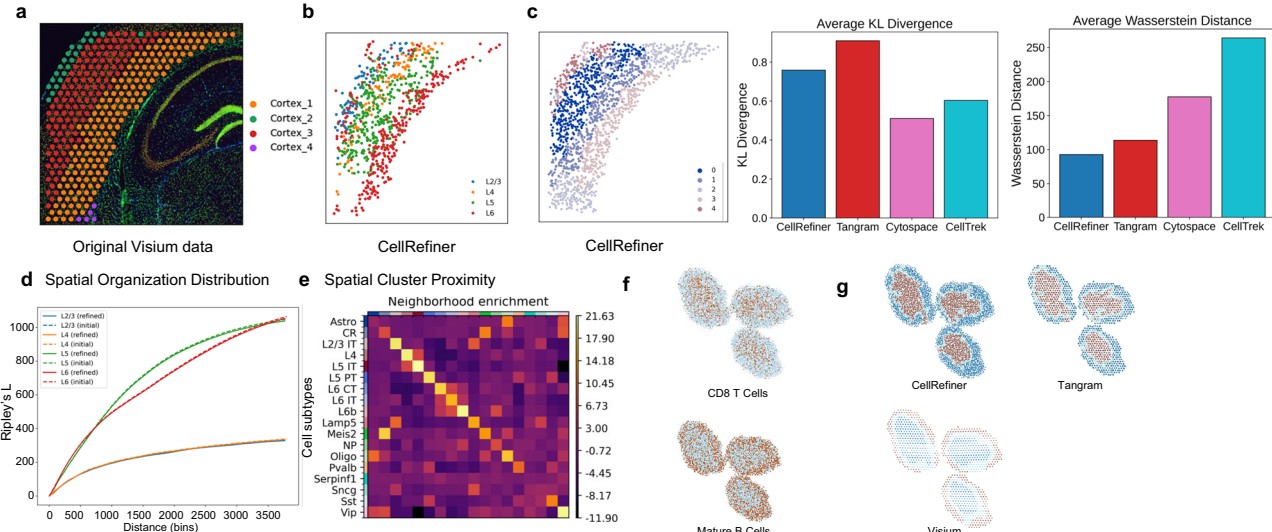

**Fig. 3 | Performance of CellRefiner on multi-cell resolution data. a** Visium ST data from a slice of the mouse cortex. **b** The distribution of annotated cell types from the scRNA-seq data on the CellRefiner output. **c** Segmentation of CellRefiner reconstruction of mouse cortex Visium data, using SpaceFlow, with colors corresponding to clusters. **d** Ripley's L for spatial organization on Visium and CellRefiner. **e** Neighborhood enrichment score on spatial proximity of clusters for CellRefiner output. **f** ST data from murine lymph node using Visium. Source data are provided as a Source Data file.

agrees well with the prior knowledge on spatial arrangements of cell types (Fig. 4c) and it is used for detailed contact-based signaling analysis, which is infeasible using the original Visium data. We then evaluate the performance of CellRefiner on contact-based cell-cell communication analysis (Supplementary Fig. 5). We compared CellRefiner with four approaches: the spatial mapping of cells generated by Tangram or CellTrek and the contact map generated with a k-nearest neighbor graph or a Delaunay graph. In the task of predicting the contact-based cell neighborhood cell type composition and contact-based signaling, CellRefiner and Tangram+Delaunay perform the best. CellRefiner achieves comparable correlation coefficients similar to the Tangram+Delaunay approach, however, with significantly lower RMSE, indicating a lower false positive rate (Supplementary Fig. 5). However, like other CCC inference methods, CellRefiner may still produce false positives and needs to be used primarily as a tool for hypothesis generation.

Based on the cell shapes modeled by CellRefiner, we analyze contact-based cell-cell communication in the multicell resolution Visium data. First, we apply CellRefiner to a Visium data of human squamous cell carcinoma (Fig. 5). We successfully identify various signaling patterns in agreement with prior knowledge (Fig. 5a) including 1) enriched EPHB signaling on the tumor region border known as a key interaction between tumor keratinocytes and the tumor microenvironment[46], 2) enriched NOTCH signaling within the tumor region known to promote angiogenesis[47], 3) decreased activity level of ICAM signaling in tumor region consistent with experimental observation[48], and 4) CDH (E-Cadherin) mediated cell-cell adhesion within the tumor region. In addition, we identify various enriched signaling in the tumor region that contribute to cell–cell adhesion and desmosome formation[49] (Fig. 5b).

Second, we analyze contact-based cell-cell communication in a Visium data of mouse cortex (Fig. 6). 1) NOTCH signaling has previously been observed in adult mouse brain with various potential functions such as neural stem cell maintenance and cell cycle maintenance[50]. Here, we observe significant NOTCH signaling across cortex layers implying its potential role in regulating global functions. 2) CDH is known to mediate cell-cell interactions that guide neuronal movement and organization[51]. In addition to enriched CDH signaling within and across cortical layers, we also identify significant CDH signaling between the cortical layers and the Sst interneurons, which may

suggest the role of CDH on wiring this interneuron type into the cortical circuit. 3) EPHA and EPHB signaling are observed both among the cortical layers and between the cortical layers and the various interneurons. Specifically, a strong signaling directionality is identified from the various interneurons to the cortical layers, indicating its various roles including neurogenesis and interneuron migration[52,53]. 4) Significant NRXN and SEMA4 signaling activity is observed within the cortical layers, consistent with their known roles in shaping neuronal synapses[54,55].

ST data at spot-level resolution assigns a single expression level to each spot and gene, while each spot often contains multiple cells. This spot-level mixing can obscure heterogeneous signaling activity within the same spot, particularly for contact-based communication. In the analysis of NOTCH signaling, several Visium spots were detected with NOTCH signaling, however CellRefiner reconstruction revealed that only a subset of the constituent cells receiving signal, indicating the presence of both NOTCH-responsive and non-responsive cells within individual spots. These results demonstrate that CellRefiner can handle the spot-level mixing and recover the heterogeneity of contact-based cell–cell communication that cannot be resolved at the spot level (Fig. 6c).

## Discussion

We developed CellRefiner to generate single-cell resolution spatial data from the paired scRNA-seq and ST data using a particle-based method combining spatial, gene expression information, and ligand-receptor interactions information. With more accurate single-cell resolution, the reconstructed data allows more and better spatial analysis of gene expression, such as cell-cell communication inference by utilizing more realistically spatial context as well as higher resolution expression information. This particle-based method is both interpretable, with an explicit formulation, and flexible, enabling its integration with a variety of other mapping methods and datasets for improvements.

We validated and benchmarked CellRefiner first on the particle-based method using pseudo-Visium datasets then evaluated the entire method on a range of different datasets such as MERFISH, seqFISH, Slide-seqV2, and STARmap. While CellRefiner generally produces accurate spatial mappings of single cells, it may not capture subtle variations in cell density across different spatial regions. This limitation

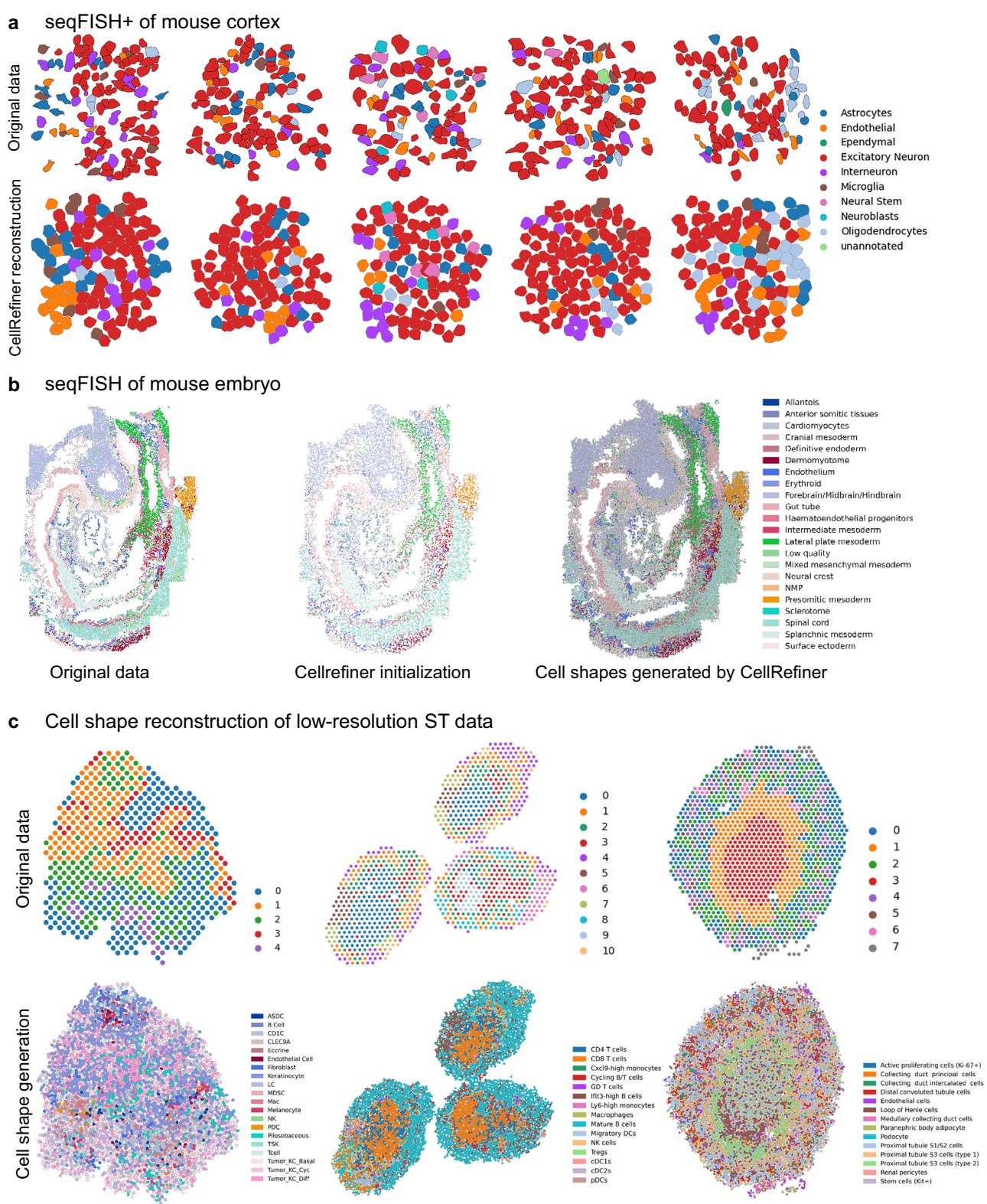

**Fig. 4 | Cell shape reconstruction using CellRefiner with each cell modeled with multiple elements. a** The original seqFISH+ mouse cortex data and the cell shape reconstruction by CellRefiner. **b** The original seqFISH mouse embryo data and the cell shape reconstruction by CellRefiner. **c** The cell shape prediction by CellRefiner on multi-cell resolution spatial data using paired scRNA-seq data of human squamous cell carcinoma, mouse lymph node, and mouse kidney. Source data are provided as a Source Data file.

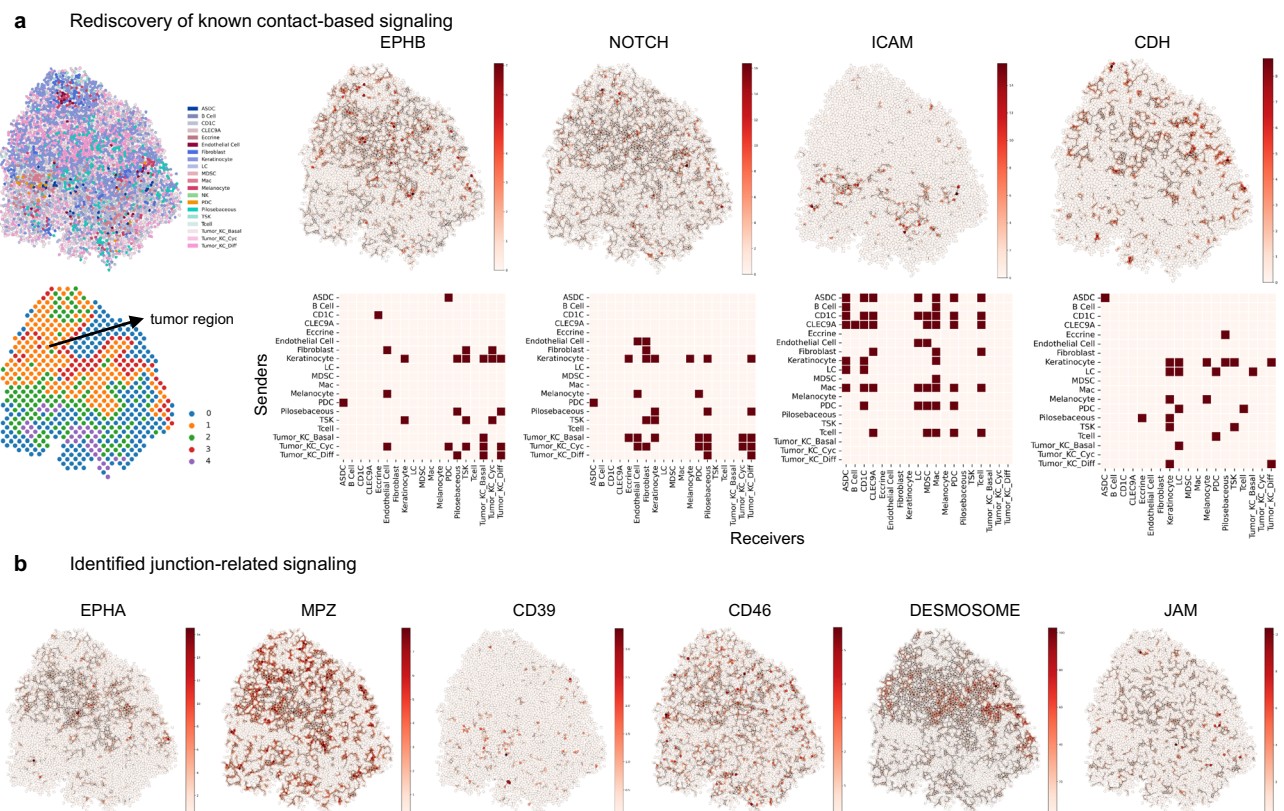

**Fig. 5 | Contact-based cell-cell communication analysis of human squamous cell carcinoma. a** The single-cell resolution spatial map of cells reconstructed by CellRefiner using a paired Visium data and scRNA-seq data. The analysis rediscovered several contact-based signaling activities confirmed by prior knowledge, including EPHB, NOTCH, ICAM, and CDH. **b** CellRefiner also identified several highly active junction-related signaling including EPHA, MPZ, CD39, CD46, DES-MOSOME, and JAM. Source data are provided as a Source Data file.

should be considered when applying it to precise physical models, such as those involving the extracellular matrix. Additionally, similar patterns are observed on the boundaries of datasets especially those with fine details (Fig. 2e, seqFISH example), which is due to the unknown boundary lines in the pseudo-Visium data. To further evaluate the spatial mapping, we have focused on the ability of CellRefiner to generate insights for CCC using contact maps that explore the higher fidelity spatial structure provided by CellRefiner over ST data. This exploration was extended to study two Visium datasets, where it again shows noticeable value over both ST data and a direct mapping method. Moreover, CellRefiner accurately recapitulates the CCC analysis from single-cell resolution spatial data as validated on seqFISH+ data while enabling new insight into CCC using Visium data. Finally, CellRefiner is computationally efficient, making it practical to apply to large-scale single-cell and spatial transcriptomics datasets (Supplementary Fig. 6).

While CellRefiner bridges an important gap between scRNA-seq and ST data, its effectiveness may be limited in certain ways. The cell-to-spot mapping is currently highly dependent on the quality of the paired datasets and needs better indication on the accuracy of the mapping. This can be countered by integrating multiple datasets and using other computational methods to correct for the potential batch effects. Also, CellRefiner assumes some homogeneity across cell sizes and densities. However, in some tissues the number of cells per spot varies or even has cavities absent of any cells, so more sophisticated modeling approaches are needed, or one needs to select datasets with relatively uniform cell densities for better accuracy when using Cell-Refiner. CellRefiner also addresses low-abundance cell types by explicitly including an option to map specified cell types. Following the initial global mapping of cellular populations to spatial spots, user-specified low-abundance cell types are then mapped to spots using gene expression similarity. This targeted allocation prevents the signals of low-abundance cells from being overshadowed by more abundant neighboring populations. The spatial refinement then optimizes the position of all the mapped cells (Supplementary Fig. 7). We also evaluated the effect of incorporating ligand-receptor-derived forces and found that they contribute only minimal refinement to the final mapping, producing reconstructions nearly indistinguishable from the results without these terms (Supplementary Fig. 8), suggesting that most of the spatial signal is already captured by the expression- and geometry-based components.

The particle-based model aspect of CellRefiner makes it a natural fit for directly implementing additional modalities to generate single-cell resolution spatial epigenomic data[56] or spatial proteomic data[57]. Another avenue for improvement is to include cell segmentation masks for imposing variable spatial constraints and more heterogeneous cell density to handle a wider range of spatial configurations. More complex forms of modeling ligand-receptor interactions or intracellular gene regulatory behavior may also further enhance the performance of CellRefiner.

This flexible framework can accommodate additional information for improvement, such as using integrated single-cell multi-omics or incorporating morphological regularizations from staining images that can be implemented as additional forces in the physical model. CellRefiner can also be applied to improved spatial transcriptomics datasets generated using methods that incorporate paired imaging data. While in this study CellRefiner is used as a data analysis method for generating single-cell resolution ST data, it can be extended as a modeling framework. For example, gene network models can be attached to each cell in CellRefiner to construct a spatiotemporal

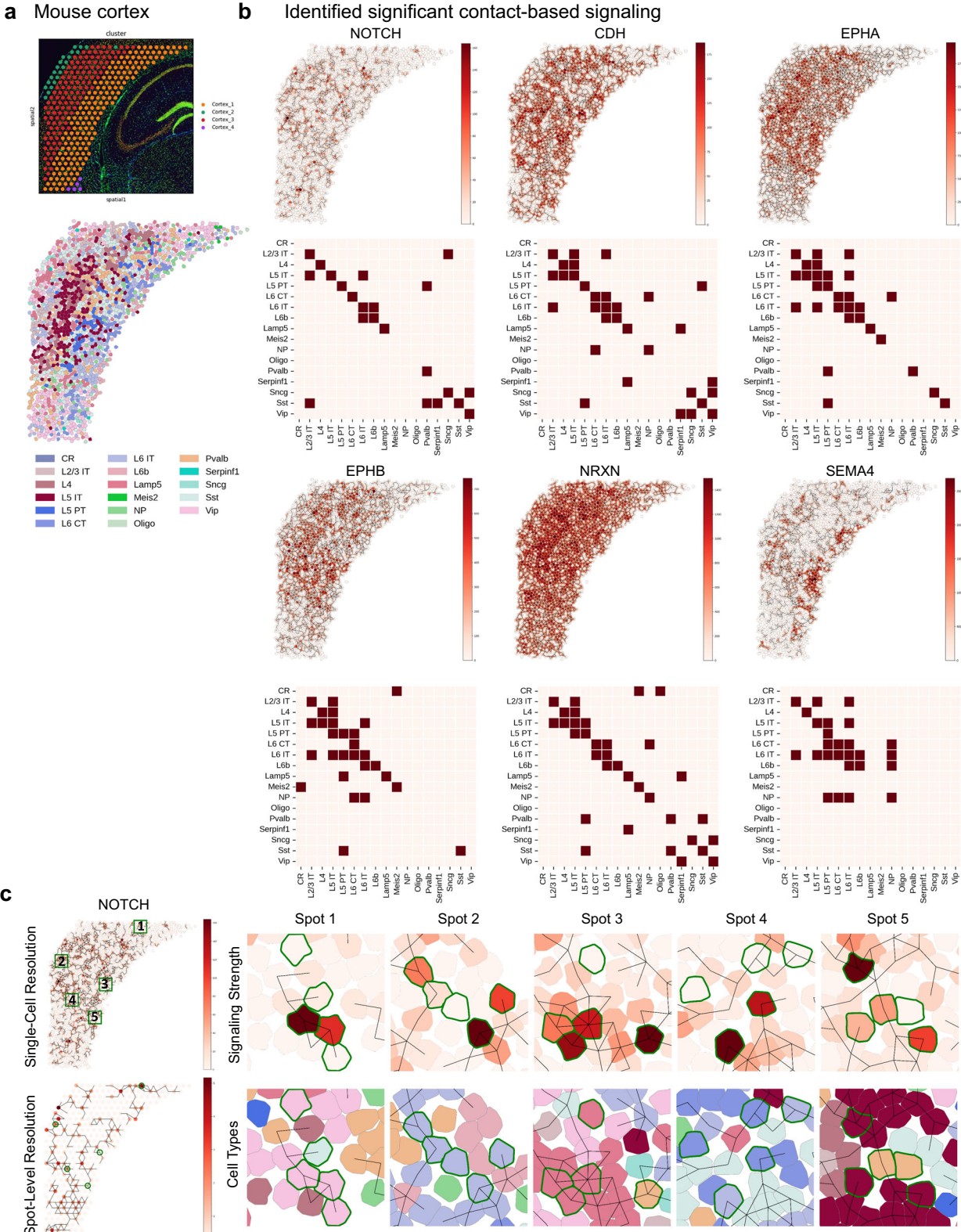

**Fig. 6 | Contact-based cell-cell communication analysis of mouse cortex. a** The single-cell resolution spatial map of cells by CellRefiner colored by the expert annotation of the scRNA-seq data. **b** The identified highly active contact-based signaling by CellRefiner, including NOTCH, CDH, EPHA, EPHB, NRXN, and SEMA4. **c** Single-cell–resolution NOTCH signaling reconstructed by CellRefiner (top) compared with spot-level NOTCH signaling inferred from Visium spot data (bottom). Five representative spots (green boxes) are shown in zoom-in panels. Although NOTCH signaling is detected at the spot level, CellRefiner reveals that only a subset of the cells within the spot receive NOTCH signaling. Green cell boundaries indicate the cells belonging to the shown spot. Source data are provided as a Source Data file.

model driven by the spatial data. Together, CellRefiner explores the usage of physical model for single-cell and spatial transcriptomics data analysis, and there are broadly other potentials beyond the application described here.

## Method
### Data preprocessing
For the scRNA-seq data, we performed a standard data analysis procedure of cell and gene filtering (filtering out cells with two or fewer genes followed by filtering out genes detected in two or fewer cells), normalizing total count, log1p normalization. A similar procedure was used for the ST data, then the two datasets were reduced to shared genes. For the scRNA-seq data, we then performed PCA and retained the first 50 components, then found the k-nearest neighbor graph. Differentially expressed genes were identified by a statistical $t$-test using the most granular pre-annotated cell type (or Leiden clustering if not pre-annotated) and the top 100 marker genes for each cell type were used as training genes for the mapping. For contact-based signaling analysis, the ligand and receptor genes are filtered by removing the genes detected in less than 1% of cells.

### CellRefiner model
CellRefiner uses a modular approach to combine sRNA-seq and ST data to generate single-cell resolution spatial data. Starting with gene expression data from scRNA-seq and ST data of the same tissue and spatial coordinates from the ST data, we find a mapping that assigns cells to spots. This mapping $\mathbf{T} \in \mathbb{R}^{c \times s}$ is generated from the Fused Gromov-Wasserstein transport equation $L(\mathbf{C}_1, \mathbf{C}_2, \mathbf{M}, \mathbf{p}, \mathbf{q})$ in Eq. (1). The cost of mapping $\mathbf{M} \in \mathbb{R}^{c \times s}$ between domains is based on the cosine dissimilarity between the ST gene expression matrix $\mathbf{G} \in \mathbb{R}^{s \times g}$ and the scRNA-seq gene expression matrix $\mathbf{S} \in \mathbb{R}^{c \times g}$, where $c$ is the number of cells, $s$ is the number of spots, and $g$ is the number of shared genes between ST data and marker genes of scRNA-seq dataset. The first cost matrix $\mathbf{C}_1 \in \mathbb{R}^{c \times c}$ is the cosine dissimilarity of gene expressions while the second cost matrix $\mathbf{C}_2 \in \mathbb{R}^{s \times s}$ aims to maximize spots that are spatially closer and have more similar gene expressions. These factors in the Euclidean distance between spot coordinates $\mathbf{A}_1 \in \mathbb{R}^{s \times s}$ (normalized to maximum of 1) as well as $\mathbf{A}_2 \in \mathbb{R}^{s \times s}$, the cosine dissimilarity of a low-dimensional embedding between spots. The weight between the two costs $\eta$ is 0.5. Here we use SpaceFlow to generate a spatially consistent low-dimensional embedding that incorporates both spatial information and gene expression. We choose $\mathbf{p}$ and $\mathbf{q}$ to be uniform distributions over the source and target domains.

$$L(\mathbf{C}_1, \mathbf{C}_2, \mathbf{M}, \mathbf{p}, \mathbf{q}) = \min_{\mathbf{T} \in \Pi(\mathbf{p}, \mathbf{q})} (1 - \eta) \sum_{i,j} \mathrm{T}_{ij} M_{ij}$$
$$+ \eta \sum_{i,j,k,l} |C_{1,ik} - C_{2,jl}|^2 \mathrm{T}_{ij} \mathrm{T}_{kl}$$

$$\mathbf{M} = 1 - \frac{\mathbf{S} \cdot \mathbf{G}}{||\mathbf{S}||_2 ||\mathbf{G}||_2}$$

$$\mathbf{C}_2 = \mathbf{A}_1 \odot \mathbf{A}_2 \quad (1)$$

This is solved using the conditional gradient to find a probability matrix, from which we take the top-$k$ cells for each spot (depending on sample and ST data collection method). The assigned cells are randomly initialized around the known spot coordinates from the ST data to initialize cell $i's$ coordinates $\mathbf{x}_i$. The spatial locations of cells are refined with a combination of several forces,

$$\dot{\mathbf{x}}_i = \mathbf{F}_{m,i}(\mathbf{x}_i) + \mathbf{F}_{s,i}(\mathbf{x}_i) + \mathbf{F}_{g,i}(\mathbf{x}_i) + \mathbf{F}_{LR,i}(\mathbf{x}_i) \quad (2)$$

These four forces acting on each cell's spatial coordinates are based on the Morse potential, spatial constraint, gene expression, and ligand-receptor interactions. The cells' locations governed by the ODE system described in Eq. (2) are then solved using a forward Euler method with fixed step size with user input $h$ (time step size default to 20) where at each iteration $t$, all $n$ cells have their cell spatial coordinates $\mathbf{x}_{t,i} \in \mathbb{R}^2$ simultaneously updated.

### The potential force calculation
The Morse potential is the basic elastic response function $\mathbf{F}_m$ in Eq. (3) to intercellular biomechanical forces acting between every pair of cells with the goal of spreading the cells out uniformly[58,59]. As the simulation progresses, the strength of this force decreases using a simulated annealing approach. Here $V$ is the Morse potential function of distance with parameters $V_0, U_0, c_1, c_2$ chosen to achieve a specific potential energy function shape (Supplementary Fig. 9). This shape achieves smoother convergence as well as a potential energy well at the desired distance between cells for optimal spacing.

$$\mathbf{F}_m(\mathbf{x}_i) = -\Sigma_{j \neq i}^N \nabla_{\mathbf{x}_i} V(\mathbf{x}_i, \mathbf{x}_j)$$

$$V(\mathbf{x}_i, \mathbf{x}_j) = V_0 \exp(-||\mathbf{x}_i - \mathbf{x}_j||^2/c_1^2) - U_0 \exp(-||\mathbf{x}_i - \mathbf{x}_j||^2/c_2^2) \quad (3)$$

### Spot constraints
Each cell also experiences a force $\mathbf{F}_s$ that keeps it within its corresponding spot. This force is oriented from the center of the spot belonging to $\mathbf{x}_i$ towards $\mathbf{x}_i$ when the cell moves outside its spot radius $r_s$ in Eq. (4), where $\mathbf{x}_{s_i}$ is the spatial coordinate of the corresponding spot. The unit vector $\hat{\mathbf{r}}$ of this force points out from the center of the spot.

$$\mathbf{F}_s(\mathbf{x}_i) = -b(||\mathbf{x}_i - \mathbf{x}_{i_s}|| - r_s)^2 \frac{\mathbf{x}_i - \mathbf{x}_{i_s}}{||\mathbf{x}_i - \mathbf{x}_{i_s}||}, \text{if} ||\mathbf{x}_i - \mathbf{x}_{i_s}|| > r_s, 0 \text{ else} \quad (4)$$

The upper bound of the magnitude of $\mathbf{F}_s(\mathbf{x}_i)$ is set to 1.5 where $b = 0.05$ is the weight of force.

### Force based on gene expression similarity
Cell positions are also refined according to the gene expression force $\mathbf{F}_g$ in Eq. (5), which is proportional to the gene expression similarity between two cells at locations $\mathbf{x}_i, \mathbf{x}_j$ oriented towards each other along unit vector $\hat{\mathbf{q}} = \frac{\mathbf{x}_i - \mathbf{x}_j}{||\mathbf{x}_i - \mathbf{x}_j||}$. The gene similarity $c = \max(\mathrm{PCC}(\mathbf{g}_i, \mathbf{g}_j), 0)$ comes from the Pearson correlation between the vectors in the PCA embeddings of gene expression, $\mathbf{g}_i$ and $\mathbf{g}_j$ corresponding to cells $i$ and $j$, chosen over other metrics for noise reduction[60]. This force is only calculated between pairs of cells within the same local spatial neighborhood $M$, which contains cells within the same spot and adjacent spots. Here $a(t) = \frac{(t - t_f)^2}{20 t_f^2}$ is a temperature scheduling term that decreases the strength of the force over iterations (indexed by $t$) to the final iteration $t_f$[61].

$$\mathbf{F}_g(\mathbf{x}_i) = -\Sigma_{j \neq i}^M a(t) c(\mathbf{g}_i, \mathbf{g}_j) \hat{\mathbf{q}}. \quad (5)$$

### Force determined by ligand-receptor interactions
The ligand-receptor interactions are incorporated into the force $\mathbf{F}_{LR}$ using a cell-cell affinity partial correlation matrix $\mathbf{W} \in \mathbb{R}^{n \times n}$ in Eq. (6) for cells within the same local neighborhood as the gene expression. From the single-cell expression matrix we define ligand and receptor expression matrices $\mathbf{T}_L$ and $\mathbf{T}_R \in \mathbb{R}^{n \times n_{LR}}$ by taking the single-cell expression profiles and subsetting them to the corresponding ligand

and receptor expressions based on $n_{LR}$ ligand-receptor pairs drawn from the CellChatDB[62]. The product of these matrices gives the affinity between pairs of cells, and since cells can express both ligand and receptor genes at the same time, we get the following affinity matrix[16].

$$\mathbf{W}_0 = \mathbf{T}_L \mathbf{T}_R^T + \mathbf{T}_R \mathbf{T}_L^T. \tag{6}$$

$\mathbf{W}_0$ is then approximated as $\mathbf{W}$ using the Spielman-Srivastava algorithm for spectral sparsification[16,63] with a threshold (default = 0.4) to refine the affinity matrix while preserving its structure, characterized as the spectrum of the graph Laplacian. The ligand-receptor force is determined similarly to the gene expression force with force between two cells at the at locations $\mathbf{x}_i$, $\mathbf{x}_j$ oriented towards each other along unit vector $\hat{\mathbf{q}} = \frac{\mathbf{x}_i - \mathbf{x}_j}{\|\mathbf{x}_i - \mathbf{x}_j\|}$ and proportional to the corresponding affinity term $w_{ij}$ given by Eq. (6).

$$\mathbf{F}_{LR}(\mathbf{x}_i) = -\Sigma_{j \neq i}^n a(t) w_{ij} \hat{\mathbf{q}}. \tag{7}$$

### Tissue boundary

We impose a hard constraint on the boundary of the tissue to prevent cells from moving outside (Supplementary Fig. 10). The boundary $x_B$ is defined by fitting a multivariate Gaussian distribution $Z(\mathbf{x}) = \Sigma_{i=1}^n N(\mathbf{x}; \mathbf{x}_i, 10^4 I_2)$ over the spatial coordinates of the initial cell mapping and taking the level set $\{\mathbf{x}_B | Z(\mathbf{x}_B) = 0.4 \max_x Z(\mathbf{x})\}$. If a cell goes outside the boundary, i.e. $Z(\mathbf{x}_i) < Z(\mathbf{x}_B)$ then it is relocated to the nearest point on the boundary.

### Cell shape modeling

A model based on the subcellular element method[26] was leveraged to reconstruct individual cell shapes from the spatial transcriptomics dataset. Each cell $i$ ($i = 1, \cdots, N$) is represented by a collection of $n_e$ elements (points) in space $\{\mathbf{x}_{ik}\}_{k=1}^{n_e}$, where $\mathbf{x}_{ik}$ is the coordinate of element $k$ and $n_e = 20$. $N$ is the total number of cells. The position and shape of each cell are captured by its collection of elements. Cell shape is driven by intercellular and intracellular physical interactions which in our model are characterized by potential between elements. The movement of element $k$ is given by an overdamped Langevin equation:

$$\begin{aligned}\frac{d\mathbf{x}_{ik}}{dt} = & -\nabla_{ik} \sum_j \sum_l \alpha_{ij} V_{\text{inter}}(\|\mathbf{x}_{ik} - \mathbf{x}_{jl}\|) \\ & -\nabla_{ik} \sum_{l \neq k} [V_{\text{intra}}(\|\mathbf{x}_{ik} - \mathbf{x}_{il}\|)] + \boldsymbol{\xi}_{ik}(t),\end{aligned} \tag{8}$$

$V_{\text{inter}}$ and $V_{\text{intra}}$ are the potential between elements, denoted intercellular and intracellular respectively. The intercellular potential $V_{\text{inter}}$ employ Lennard-Jones form to model the adhesion and repulsion between cells, preventing cells from overlapping while promoting the formation of physical contacts between neighboring cells.

$$V_{\text{inter}}(r) = \left(\frac{r_{\text{inter}}}{r}\right)^{12} - 2\left(\frac{r_{\text{inter}}}{r}\right)^6, \tag{9}$$

where $r$ is the distance between elements and $r_{\text{inter}} = 2.4$ denotes the equilibrium distance at which the potential is minimized. The parameter $\alpha_{ij}$ is the interaction strength between cell $i$ and cell $j$, derived from the gene expression similarity. $\alpha_{ij}$ is calculated as $\max(\text{PCC}(\mathbf{g}_i, \mathbf{g}_j), 0.05)$ and then scaled to $[0.04, 1]$, where $\text{PCC}(\mathbf{g}_i, \mathbf{g}_j)$ is the Pearson correlation between the PCA embeddings of gene expression $\mathbf{g}_i$ and $\mathbf{g}_j$ for cells $i$ and $j$. The intracellular potential $V_{\text{intra}}$

ensures the integrity of cell shapes.

$$V_{\text{intra}}(r) = \beta \left[ \left(\frac{r_{\text{intra}}}{r}\right)^{12} - 2\left(\frac{r_{\text{intra}}}{r}\right)^6 \right] + \gamma r^3. \tag{10}$$

The first term is a Lennard-Jones potential with $r_{\text{intra}} = 2$ denoting the equilibrium distance, maintaining uniform spacing of elements and cell volume conservation with $\beta = 0.25$. The second term is a penalty term that grows rapidly at large distances and penalizes excessive stretching of the element network with $\gamma = 0.001$, preventing highly deformed or fragmented shapes. $\boldsymbol{\xi}_{ik}$ is the Gaussian white noise with zero mean and correlation $\langle \xi_{ik}^m(t), \xi_{jl}^n(t') \rangle = \sigma^2(t) \delta(t - t') \delta_{m,n} \delta_{i,j} \delta_{k,l}$, where $\sigma(t)$ is noise strength, $m$ and $n$ are indices of spatial dimension, $\delta$ is Kronecker delta function. To allow the system to converge to stable cell shapes, the noise strength decays via a simulated annealing schedule $\sigma(t) = \sqrt{(T_{\max} - t)/T_{\max}}$, where $0 \leq t \leq T_{\max}$ and $T_{\max}$ is the final integration time.

Spatial coordinates of cells were extracted from the spatial transcriptomics dataset and processed to serve as the cell centroids. The median intercellular distance $d_m$ was calculated using Delaunay triangulation and the empirical cell radius was estimated as $d_m/2$. Because the steady-state radius of a cell in the subcellular element model is approximately $\sqrt{n_e} r_{\text{intra}}/2$, the spatial coordinates were scaled by $1.25 * \sqrt{n_e} r_{\text{intra}}/d_m$ to avoid generating excessively dense or sparse cell distributions. After the scaling, the initial coordinates of elements were sampled around the centroid of each cell $i$. The system then evolved according to Eq. (8). The final element coordinates $\{\mathbf{x}_{ik}(T_{\max})\}$ represent the reconstructed shape of each cell.

**Mapping of low-abundance cell types.** Low-abundance cell types may be excluded in the unbiased initial mapping due to subsampling. CellRefiner offers an option to further map user-specified cell types. Following the initial mapping $\mathbf{T}$ that assigns cells from the entire scRNA-seq dataset to spots, we additionally map the cells belonging to the specified cell types to the spatial spots. For each cell $i$ belonging to the specified cell type that is not included in the initial mapping, we assign it to spot $j^*$ that has the most similar gene expression using the cosine similarity metric: $j^* = \text{argmax}_j \frac{\mathbf{S}_i \cdot \mathbf{G}_j}{|\mathbf{S}_i||\mathbf{G}_j|}$. The set of cells from both the initial mapping and the specified cell types is then passed to the spatial refinement step of CellRefiner.

**Pseudo-Visium data generation.** For each dataset, we create a uniform grid such that each grid point has at least $n = 5$ cells. Using a KD-tree, cells are then assigned by distance to grid points to a maximum of $n$ cells per grid point, and within half the distance to the neighboring grid point. A spot is then defined as the spatial coordinates of the grid point and the aggregate gene expression of the cells assigned to the grid point. Spots without any cells assigned to them are removed. For the MERFISH dataset, a 32 spot by 32 spot grid was created with a radius of 0.016. For the seqFISH dataset of mouse embryo, a 62 spot by 62 spot grid was created with a radius of 0.16, then spots with no cells within the radius were removed. For the Slide-seqV2 dataset, a 28 spot by 28 spot grid was created with an 80 micron radius spot size, and empty spots removed. For the STARmap dataset, a 24 spot by 10 spot grid was created with a 300 micron radius. For the seqFISH+ dataset of mouse cortex, a 35 spot by 5 spot grid was created with a radius of 187 micron radius spot size, then spots with no cells within the radius were removed.

**Contact-based signaling analysis.** Contact-based signaling is transduced via ligand-receptor interactions at the interfaces of cell-cell physical contacts, which can be delineated by subcellular element representation. To determine the physical contacts between cells, a

k-nearest neighbor (k = 8) graph between all elements is constructed. If one element of cell $j$ appeared among the nearest neighbors of any elements of cell $i$, the two cells were considered to be in physical contact. Summarizing the neighbors of elements of each cell yields a set of undirected cell-cell contacts $C = \{(i,j)\}$. For each contacting cell pairs $(i,j) \in C$, we quantified the potential contact-based cell-cell communication mediated by every ligand–receptor pairs using a communication score.

$$\mathbf{s}_{ij}^{LR} = \mathbf{g}_i^L \mathbf{g}_j^R \tag{11}$$

$\mathbf{g}_i^L$ and $\mathbf{g}_j^R$ denote the normalized gene expressions of ligand L in cell $i$ and receptor R in cell $j$. The ligand-receptor pairs are identified using CellChatDB[62].

**Cluster-level communication.** The cluster-cluster communication strength for each ligand–receptor pair is computed as mean of all cell-cell communication scores between the two clusters $\mathbf{S}_{IJ}^{LR} = \sum_{i \in I, j \in J} \mathbf{s}_{ij}^{LR} / (N_I N_J)$. The significance of cluster-cluster communication of each L-R pair is determined by a permutation test. We generated null distribution by $n$ independent permutations of the cell-to-cluster assignment and recalculating the cluster-cluster communication scores ($n = 100$ by default). The $p$-value for cluster-cluster communication is given by the percentile of the original $\mathbf{S}_{IJ}^{LR}$ within the null distribution. Communications with $p$-value $< 0.05$ are considered significant.

**Optimization.** The optimal transport mapping was calculated using the fused_gromov_wasserstein function from the POT package with L2 loss function, and a maximum of 10,000 iterations. The spatial embedding based on the SpaceFlow implementation was run with spatial regularization strength 0.1, 0.001 learning rate, 1000 epochs, and 50 latent dimensions. The cell refinement step of CellRefiner was run with $h = 20$ and 10 iterations. Top 5 cells were assigned to each spot for pseudo ST data of MERFISH, seqFISH, STARmap and Slide-seqV2, as well as Visium data of human SCC, mouse kidney, mouse lymph node and mouse cortex. Top 4 cells were assigned to each spot for the pseudo ST data of seqFISH+ due to the small cell number in this dataset. The cell shape modeling was run with a time step of 0.04 and 2000 iterations.

**Parameter selection.** The parameters $U_0$, $V_0$, $c_1$, $c_2$ in the Morse potential (Eq. 3) were chosen such that a certain equilibrium distance between particles is achieved. The equilibrium radius was chosen as the minimum distance between spot centers multiplied by 0.35. The equilibrium radius turned out to be 210 $\mu m$ for STARmap, 19 $\mu m$ for Slide-seqV2, 0.016 for MERFISH, and 0.041 for seqFISH. In the cell shape modeling, the cell radius scaling factor was set to 2.5 for the Visium data of human SCC, mouse kidney, mouse lymph node and mouse cortex and to 3.0 for the seqFISH+ data of mouse cortex. To achieve a target equilibrium radius in different datasets, we used the following values of $U_0$, $V_0$, $c_1$, $c_2$, STARmap: (2.68, 29.48, 32.42, 50.92); Slide-seqV2: (0.68, 7.52, 8.27, 12.98); MERFISH: ($5.61 \times 10^{-4}$, $6.18 \times 10^{-3}$, $6.79 \times 10^{-3}$, $1.07 \times 10^{-2}$); seqFISH mouse embryo: (1.39, 15.33, 16.86, 26.48); seqFISH+ mouse cortex: (2.58, 28.38, 31.22, 49.02); Visium mouse kidney: (2.34, 25.79, 28.37, 44.55); Visium mouse lymph node: (1.87, 20.60, 22.66, 35.59); Visium mouse cortex: (3.65, 40.13, 44.15, 69.32); human SCC: (3.48, 38.25, 42.08, 66.08).

**Evaluation metrics.** The KL-divergence was calculated to compare the ground truth spatial distribution of a cell type with the spatial distribution of a comparison dataset's corresponding cells, across every cell type with at least 5 cells. Given two 2D point clouds corresponding to the respective spatial distributions of the cells, we create a 100 by 100 grid over the region bounded by the min and max of the point clouds. We then evaluate the densities $p(x)$ and $q(x)$ as histograms at the grid points $x_i$. The densities are normalized, from which we then calculate the KL divergence:

$$KL(p, q) = \sum_i p(x_i) \log\left(\frac{p(x_i) + \epsilon}{q(x_i) + \epsilon}\right) \tag{12}$$

The Wasserstein metric is used similarly for each pair of cell types with least 5 cells. First, we compute a cost matrix using the Euclidean distance between the ground truth cells' spatial coordinates and the comparison cells' spatial coordinates. Then find the Wasserstein distance using the emd2 function from the POT toolbox[64], which solves the earth movers distance problem with uniform weights. For comparing spatial distributions, we implement a Ripley's L-function $L(r)$ for each cell type in the dataset by estimating the K-function $K(r)$ and transforming it. $a$ is the area of the region, $n$ the number of points, $I$ the indicator function and $d_{ij}$ the distance between two points. The max radius $r$ is chosen by taking a quarter of the minimum dimension of the bounding box containing the spatial region. The mean squared error over 20 bins is used as the evaluation metric.

$$K(r) = \frac{a}{n(n-1)} \sum_i \sum_j I\left(d_{ij} \le r\right)$$

$$L(r) = \sqrt{\left(\frac{K(r)}{\pi}\right)} \tag{13}$$

Neighborhood enrichment score was calculated using Squidpy[42] to find clusters that are neighbors in the tissue of interest. This is done by generating a spatial connectivity graph across cells and observing the number of edges between different clusters. A permutation test on the cluster labels is performed with $n = 1000$ to create a distribution of expected counts under the null hypothesis of no spatial enrichment. We compute a z-score between each pair of clusters to get the neighborhood enrichment score, with a high z-score indicating significant enrichment.

The perturbed MERFISH data used for evaluation was generated using Gaussian noise with sigma 0.02. CellRefiner was run for 10 iterations with time step 0.5 and hyperparameter $m = 0.025$.

For benchmarking, CytoSPACE was run using 5 cells per spot and default parameters otherwise, with a spatial location radius of 0.016 on seqFISH and MERFISH datasets, 300 microns for STARmap, and 80 microns for Slide-seqV2. CellTrek was run with PCA reduction to 30 components, nonlinear interpolation with 5000 points, 1000 tree random forest, 0.55 distance threshold, max of 5 spots per cell, (spot_$n = 5$, repel_r = 20, repel_iter=20). Tangram was run with default settings and 500 epochs for training.

**Reporting summary**
Further information on research design is available in the Nature Portfolio Reporting Summary linked to this article.

**Data availability**
The mouse hypothalamic preoptic region MERFISH original data is available at https://doi.org/10.5061/dryad.8t8s248[35] with pre-processed data available via the Squidpy package[42]. The mouse visual cortex STARmap data[29] is accessible via the STAGATE[65] package. The mouse hippocampus Slide-seqV2 data[10] is available at the Broad Institute Single Cell Portal (https://singlecell.broadinstitute.org/single_cell/study/SCP815/sensitive-spatial-genome-wide-expression-profiling-at-cellular-resolution#study-summary) with preprocessed data available via the Squidpy package[42]. The preprocessed seqFISH data of mouse embryo[37] is available via the Squidpy package[42]. The human breast cancer Xenium data[38] is available at the 10X Genomics

website (https://www.10genomics.com/products/xenium-in-situ/preview-dataset-human-breast). The mouse brain (sagittal posterior) Visium data is available at the 10X Genomics website (https://www.10genomics.com/resources/datasets/mouse-brain-serial-section-1-sagittal-anterior-1-standard-1-1-0). Corresponding scRNA-seq data[39] is available at the Gene Expression Omnibus (GEO, https://www.ncbi.nlm.nih.gov/geo) under accession code GSE115746. Both 10X Visium and 10X Chromium samples[43] for the lymph node are available at https://github.com/romain-lopez/DestVI-reproducibility. Human SCC Visium data and scRNA-seq data[46] are available at GEO under accession code GSE144240. seqFISH+ mouse cortex data[7] is available at https://github.com/CaiGroup/seqFISH-PLUS. The mouse kidney Visium data is available at the 10X Genomics website (https://www.10genomics.com/datasets/mouse-kidney-section-coronal-1-standard-1-1-0). The corresponding scRNA-seq data[66] is available at GEO under accession code GSE117089. The ligand-receptor pairs with secreted ligands, as categorized in the CellChatDB[62], were used and can be accessed at http://www.cellchat.org/cellchatdb/. Source data are provided with this paper.

## Code availability

All codes were implemented in Python. The package, tutorial, and reproducible code can be found at https://github.com/XiangyuKuang/cellrefiner, archived at Zenodo https://doi.org/10.5281/zenodo.18263150[67].

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

## Acknowledgements

The project was partly supported by the National Science Foundation grants DMS2151934 (Z.C.), DMS176372 (Q.N.), and CBET2134916 (Q.N.), the National Institutes of Health grants R01GM152494 (Z.C. and Q.N.), and R01AR079150 (Q.N.), and a Simons Foundation Grant 594598 (Q.N.).

## Author contributions

Z.C. and Q.N. conceived the project; E.B.C. and X.K. implemented the algorithm and code. E.B.C., X.K. and Z.C. conducted data analysis. Z.C. and Q.N. supervised the research. All the authors wrote and approved the manuscript.

## Competing interests

The authors declare no competing interests.
