## [Transparent Peer Review file · Nature Communications]

Reconstructing single-cell resolution from spatial transcriptomics with CellRefiner

Corresponding Author: Professor Qing Nie

Version 0:

Reviewer comments:

Reviewer #2

(Remarks to the Author)

The authors have made substantial revisions and addressed my major concerns from the first review. The manuscript has been significantly improved in both methodology and clarity. I only have two minor comments for further refinement:

1. As shown in Supplementary Fig. 3, the performance of CellRefiner decreases on rare cell types (<5% of the population). While the authors acknowledge this limitation as a caveat, no strategy is provided to mitigate the issue. This could be an important pain point in real applications, especially for studies focusing on rare immune or stem cell populations. A brief discussion on potential future directions to handle rare cell types would strengthen the manuscript.

2. The font size in the main figures is not uniform, which affects readability. For example, the bar plots in Fig. 2d and Fig. 3c use noticeably smaller fonts compared to other panels. I recommend revising the figure texts to ensure consistency with journal formatting requirements.

(Remarks on code availability)

The reproducibility of the figures is very good.

Reviewer #3

(Remarks to the Author)

I was asked to advise whether the authors have appropriately addressed the concerns of Reviewer #1.

I find the comments of reviewer #1 reasonable, focusing mainly on two aspects. First, demonstrating biological relevance of task addressed by the method and the interpretability of the results. Second, addressing the limitations related to applicability and false positive results.

The authors have made an effort to address the comments and have made changes to their method introducing a new aspect of their method in the direction of modeling cell shapes to address directly comment 3 of reviewer #1 and have expanded their results to demonstrate biological relevance through added results of downstream cell-cell communication analysis.

Major concerns remain about the revised version of the manuscript.

The responses to Comments 1–3 rely heavily on the new cell–cell communication analysis and the results subsection “CellRefiner provides insight into contact-based cell–cell communication.”

The identification of communication patterns appears circular. The model infers cell coordinates based on the sum of four force terms, one of which is a “force determined by ligand-receptor interactions,” calculated as a function of ligand and receptor expression pairs informed by the CellChat database. The subsequent evaluation of cell–cell communication

patterns in the reconstructed space again depends on a communication score derived from these same ligand-receptor pairs and the same database (see Methods subsection "Contact-based signaling analysis").

Since the the location is determined, in part, by optimizing for the ligand-receptor patterns, it would be surprising if CellRefiner didn't recover these same patterns. While these analysis may be used to show that CellRefiner accurately takes into account the different forces, this doesn't show the biological relevance of CellRefiner. The cell-cell communication between clusters or niches or layers can be performed also without reconstructing the single-cell resolution.

Since the inferred locations are determined, in part, by optimizing for ligand-receptor expression patterns, it is unsurprising that CellRefiner recovers those same patterns in downstream analysis. While this may demonstrate internal consistency in solving the optimization task, it does not establish biological relevance. Moreover, cell-cell communication between clusters, niches, or tissue layers can be analyzed without reconstructing single-cell spatial positions.

Comment 4 has been addressed.

Regarding Comment 5, I would not consider a direct comparison to CeLEry strictly necessary in the absence of a relevant downstream task, provided the tool cannot be reasonably executed. However, if comparison is deemed essential, it could be performed on a subset of the dataset used in the CeLEry publication, where results should be reproducible.

The responses to Comments 6 and 7 are more concerning. The authors seem to confirm that CellRefiner reconstructs only the positions of abundant cell types only, without taking additional steps to address these comments. This limitation again raises questions about the biological relevance of the reconstructed spatial structure and the potential for meaningful downstream analyses.

Comment 8 was addressed, but the reported runtime scaling is unexpected. It appears to increase sub-linearly with the number of cells (1000 - ~20s, 2000 - ~40s, 4000 - ~60s, 7000 - ~80s) This is inconsistent with the stated linear time and space complexity of the algorithm. Furthermore, the dataset sizes used are too small to convincingly demonstrate scalability. A single Visium slide contains approximately 5000 spots, each capturing 1-10 cells. The authors should therefore scale up their analysis and explicitly report the hardware configuration and any paralelization or other acceleration used for runtime evaluation.

(Remarks on code availability)

Version 1:

Reviewer comments:

Reviewer #2

(Remarks to the Author)

I appreciate the authors' detailed response regarding the handling of low-abundance cell types. The addition of the optional post-hoc mapping step appears to effectively mitigate the performance drop on rare cell populations, as demonstrated by the improvements shown in the new Supplementary Figure 7.

Furthermore, the readability of the figures has been improved with consistent font sizes as requested.

I have no further comments and recommend the manuscript for publication.

(Remarks on code availability)

Reviewer #3

(Remarks to the Author)

The authors have addressed the majority of my comments. I have two remaining concerns.

1. Regarding comment 1, since the LR force term had essentially no effect on the localization and only adds to the computational complexity of the approach the authors might consider removing the term completely. The added experiments do not show a use case where this term is useful. Alternatively the observation that "incorporating LR forces leads to only marginal improvement, with performance nearly identical to that obtained without LR forces" should be clearly stated not only in the discussion but also in the results section to avoid any confusion.

2. Regarding comment 4, the authors should align the x-axes (cell-type labels) of Supplementary Figure 7 a and b. Please also check that the reported average mapping errors are correct. Looking at Supplementary Figure 7c, both spatial coordinates are in the range [0,1]. In such case the average reported error close to 0.5 for many cell types (even more frequent ones) is comparable to the expected error of randomly placed cells which is concerning. To Supplementary Figure 7b please add a baseline average mapping error based on a randomly placed cells per cell type to clarify this.

(Remarks on code availability)

Response to Reviewer 2

General comments:

The authors have made substantial revisions and addressed my major concerns from the first review. The manuscript has been significantly improved in both methodology and clarity. I only have two minor comments for further refinement.

(Remarks on code availability): The reproducibility of the figures is very good.

Response: We thank the reviewer for the constructive comments, which have helped substantially improve the manuscript. Our detailed responses to the two remaining points are listed below.

Specific comments:

1. As shown in Supplementary Fig. 3, the performance of CellRefiner decreases on rare cell types (<5% of the population). While the authors acknowledge this limitation as a caveat, no strategy is provided to mitigate the issue. This could be an important pain point in real applications, especially for studies focusing on rare immune or stem cell populations. A brief discussion on potential future directions to handle rare cell types would strengthen the manuscript.

Response: We thank the reviewer for bringing up the limitation on low-abundance cell types, which is a common challenge for computational approaches. As you mentioned, we do observe a decrease in performance of CellRefiner on low-abundance cell types. The drop in performance is because that the cells with low abundance are likely excluded in the initial mapping due to the unbiased random subsampling. In this revision, we have added an optional step to better handle these cells, by post hoc mapping user-specified cell types on top of the initial mapping. This additional treatment leads to substantial improvement in performance on these cell types (new Supplementary Fig. 7). In the revised manuscript, we have added discussion on limitations when handling low-abundance cells and added details about the optional treatment of such cells in Methods.

2. The font size in the main figures is not uniform, which affects readability. For example, the bar plots in Fig. 2d and Fig. 3c use noticeably smaller fonts compared to other panels. I recommend revising the figure texts to ensure consistency with journal formatting requirements.

Response: We thank the reviewer for raising this point. We have increased the font size in all figures.

Response to Reviewer 3

General comments:

I was asked to advise whether the authors have appropriately addressed the concerns of Reviewer #1.

I find the comments of reviewer #1 reasonable, focusing mainly on two aspects. First, demonstrating biological relevance of task addressed by the method and the interpretability of the results. Second, addressing the limitations related to applicability and false positive results.

The authors have made an effort to address the comments and have made changes to their method introducing a new aspect of their method in the direction of modeling cell shapes to address directly comment 3 of reviewer #1 and have expanded their results to demonstrate biological relevance through added results of downstream cell-cell communication analysis.

Major concerns remain about the revised version of the manuscript.

Response: We thank the reviewer for the careful evaluation of our manuscript and for acknowledging the efforts made in the previous round of revision. We value the reviewer's constructive assessment of remaining concerns. In this revision, we have carefully addressed these points. Specifically, we have resolved the concern regarding the potential circularity in CCC analysis and computational scalability. We have also provided a new module to handle low-abundance cell types. Detailed responses to these specific points are provided below.

Specific comments:

1. The responses to Comments 1–3 rely heavily on the new cell–cell communication analysis and the results subsection “CellRefiner provides insight into contact-based cell–cell communication.”

The identification of communication patterns appears circular. The model infers cell coordinates based on the sum of four force terms, one of which is a “force determined by

ligand-receptor interactions,” calculated as a function of ligand and receptor expression pairs informed by the CellChat database. The subsequent evaluation of cell–cell communication patterns in the reconstructed space again depends on a communication score derived from these same ligand-receptor pairs and the same database (see Methods subsection “Contact-based signaling analysis”).

Since the location is determined, in part, by optimizing for the ligand-receptor patterns, it would be surprising if CellRefiner didn't recover these same patterns. While these analyses may be used to show that CellRefiner accurately takes into account the different forces, this doesn't show the biological relevance of CellRefiner. The cell-cell communication between clusters or niches or layers can be performed also without reconstructing the single-cell resolution.

Since the inferred locations are determined, in part, by optimizing for ligand-receptor expression patterns, it is unsurprising that CellRefiner recovers those same patterns in downstream analysis. While this may demonstrate internal consistency in solving the optimization task, it does not establish biological relevance. Moreover, cell-cell communication between clusters, niches, or tissue layers can be analyzed without reconstructing single-cell spatial positions.

Response: We thank the reviewer for identifying this potential circularity in the method. Upon re-examining this point, we realized that the LR force was unused in the previously presented results. This was due to a singularity check of the LR force matrix which was almost always singular due to the sparsity of LR interactions. Thus, LR force was not incorporated in the combined force of the CellRefiner model. We have confirmed that all results related to CCC presented in the revised manuscript do not incorporate LR forces when performing spatial refinement, and therefore, the potential circular issue does not affect any of the reported findings.

To further clarify this point, we performed an experiment in comparing the results with and without using LR forces in the spatial placement (new Supplementary Fig. 8). We

observe that compared to results obtained without LR forces, adding LR forces only causes very minor local displacement of cells, on average, less than 20% of cell radius. Even for outliers, the displacement is at most comparable to cell radius. In evaluating contact-based CCC inference, incorporating LR forces leads to only marginal improvement, with performance nearly identical to that obtained without LR forces. We keep the LR-force as an optional module in the framework because it is conceptually important, fully compatible with the model design, and may become useful when LR interaction data is denser or more reliable.

2. Comment 4 has been addressed.

Response: We thank the reviewer for confirming that Comment 4 has been addressed.

3. Regarding Comment 5, I would not consider a direct comparison to CeLEry strictly necessary in the absence of a relevant downstream task, provided the tool cannot be reasonably executed. However, if comparison is deemed essential, it could be performed on a subset of the dataset used in the CeLEry publication, where results should be reproducible.

Response: We thank the reviewer for clarifying the suggestion about comparing to CeLEry. We would like to clarify that CeLEry is designed for the scenario with high-resolution spatial data serving as training set, while the focus of CellRefiner and other compared methods (Tangram, Cytospace, and CellTrek) is on mapping single cells from scRNA-seq to low-resolution spatial data. They therefore are designed for different application scenarios. In this revision, we have added a comparison to CeLEry using a dataset where CeLEry has been applied under the CeLEry setup (training on high-resolution ST data). When we apply CeLEry to our setup (training on low-resolution ST data and assuming the high-resolution ST data as unknown), CellRefiner significantly outperforms CeLEry. We believe this comparison clarifies the different application scenarios of the method but is not central to the main evaluation. Therefore, we have included this analysis in the Supplementary Information (new Supplementary Fig. 3).

4. The responses to Comments 6 and 7 are more concerning. The authors seem to confirm that CellRefiner reconstructs only the positions of abundant cell types only, without taking additional steps to address these comments. This limitation again raises questions about the biological relevance of the reconstructed spatial structure and the potential for meaningful downstream analyses.

Response: We thank the reviewer for raising this very important point. We have carefully evaluated the reason for the low performance on low-abundant cell types and found that it is because such cells are likely excluded in the initial mapping due to an unbiased subsampling. According to this, in this revision, we address the issue by refining the initial mapping step. Specifically, we have added an additional module to the CellRefiner workflow, that further maps user-specific cell types, particularly the low-abundant cells, to the spatial spots. As a result, we observe relative uniform performance on cells with vary abundance (new Supplementary Fig. 7). In the revised manuscript, we have added the following clarification sentence.

“CellRefiner also addresses low-abundance cell types by explicitly including an option to map specified cell types. Following the initial global mapping of cellular populations to spatial spots, user-specified low-abundance cell types are then mapped to spots using gene expression similarity. This targeted allocation prevents the signals of low-abundance cells from being overshadowed by more abundant neighboring populations. The spatial refinement then optimizes the position of all the mapped cells (Supplementary Fig. 7).”

We have also added details of the additional mapping step in the Methods section.

5. Comment 8 was addressed, but the reported runtime scaling is unexpected. It appears to increase sub-linearly with the number of cells (1000 - ~20s, 2000 - ~40s, 4000 - ~60s, 7000 - ~80s) This is inconsistent with the stated linear time and space complexity of the algorithm. Furthermore, the dataset sizes used are too small to convincingly demonstrate

scalability. A single Visium slide contains approximately 5000 spots, each capturing 1-10 cells. The authors should therefore scale up their analysis and explicitly report the hardware configuration and any parallelization or other acceleration used for runtime evaluation.

Response: We thank the reviewer for identifying this inconsistency. We apologize for the inaccuracy in the previous version where the theoretical linear complexity comes from the cell listing algorithm used by the spatial refinement part of CellRefiner. The full pipeline does not have linear scaling. In the revised manuscript, we have evaluated the computational efficiency on a wide range of number of cells, from 1,000 to 40,000, where the CellRefiner computation can still be carried out efficiently with over 40,000 cells (new Supplementary Fig. 6). This demonstrates the practical usability of CellRefiner. Due to the various components of the pipeline, the theoretical complexity analysis may be difficult to interpret for practical applications and thus we present the empirical efficiency instead.

Response to Reviewer 2

I appreciate the authors' detailed response regarding the handling of low-abundance cell types. The addition of the optional post-hoc mapping step appears to effectively mitigate the performance drop on rare cell populations, as demonstrated by the improvements shown in the new Supplementary Figure 7.

Furthermore, the readability of the figures has been improved with consistent font sizes as requested.

I have no further comments and recommend the manuscript for publication.

Response: We thank the reviewer for their positive assessment and for the constructive comments which have significantly improved the manuscript.

Response to Reviewer 3

The authors have addressed the majority of my comments. I have two remaining concerns.

Response: We thank the reviewer for the constructive suggestions. Our detailed responses to the two remaining points are provided below.

1. Regarding comment 1, since the LR force term had essentially no effect on the localization and only adds to the computational complexity of the approach the authors might consider removing the term completely. The added experiments do not show a use case where this term is useful. Alternatively the observation that "incorporating LR forces leads to only marginal improvement, with performance nearly identical to that obtained without LR forces" should be clearly stated not only in the discussion but also in the results section to avoid any confusion.

Response: We thank the reviewer for the suggestion to further clarify the point. In the revised manuscript, we have added a clarification at the place where LR force is first introduced (Results – Overview of CellRefiner): “In ablation analysis, incorporating LR forces leads to only marginal improvement, with performance nearly identical to that obtained without LR forces. We therefore treat the LR term as optional.”

2. Regarding comment 4, the authors should align the x-axes (cell-type labels) of Supplementary Figure 7 a and b. Please also check that the reported average mapping errors are correct. Looking at Supplementary Figure 7c, both spatial coordinates are in the range [0,1]. In such case the average reported error close to 0.5 for many cell types (even more frequent ones) is comparable to the expected error of randomly placed cells which is concerning. To Supplementary Figure 7b please add a baseline average mapping error based on a randomly placed cells per cell type to clarify this.

Response: We thank the reviewer for raising this important point. First, we would like to clarify the reason for the observed large error. The error was computed for each individual cell between their location in ground-truth and their mapped location, and then averaged over all cells within each cell type. This evaluation could result in large error even for high quality mapping. For example, randomly switching locations of cells of the same type would still be an accurate mapping but will lead to large errors. In the revised manuscript, we have followed the reviewer’s suggestion and added baseline errors for random mapping (Supplementary Figure 7c), which shows that while the errors are relatively large, they are much lower than random baselines.

Additionally, to mitigate the limitation of unreasonably large error due to assessing individual cells, we have added an evaluation using Wasserstein distance between the ground-truth distribution and the mapped distribution of each cell type. This new metric shows much smaller error (<0.1 for abundant cell types and mostly <0.2 for less abundant cell types).

We have also aligned the x-axes for the three panels (a-c) in the revised Supplementary Figure 7.